# Augmented Cyclic Adversarial Learning for Low Resource Domain Adaptation

**Ehsan Hosseini-Asl, Yingbo Zhou, Caiming Xiong, Richard Socher**
Salesforce Research
{ehosseiniasl,yingbo.zhou,cxiong,rsocher}@salesforce.com

## Abstract

Training a model to perform a task typically requires a large amount of data from the domains in which the task will be applied. However, it is often the case that data are abundant in some domains but scarce in others. Domain adaptation deals with the challenge of adapting a model trained from a data-rich source domain to perform well in a data-poor target domain. In general, this requires learning plausible mappings between domains. CycleGAN is a powerful framework that efficiently learns to map inputs from one domain to another using adversarial training and a cycle-consistency constraint. However, the conventional approach of enforcing cycle-consistency via reconstruction may be overly restrictive in cases where one or more domains have limited training data. In this paper, we propose an augmented cyclic adversarial learning model that enforces the cycle-consistency constraint via an external task specific model, which encourages the preservation of task-relevant *content* as opposed to exact reconstruction. We explore digit classification in a low-resource setting in supervised, semi and unsupervised situation, as well as high resource unsupervised. In low-resource supervised setting, the results show that our approach improves absolute performance by $14\%$ and $4\%$ when adapting SVHN to MNIST and vice versa, respectively, which outperforms unsupervised domain adaptation methods that require high-resource unlabeled target domain. Moreover, using only few unsupervised target data, our approach can still outperforms many high-resource unsupervised models. Our model also outperforms on USPS to MNIST and synthetic digit to SVHN for high resource unsupervised adaptation. In speech domains, we similarly adopt a speech recognition model from each domain as the task specific model. Our approach improves absolute performance of speech recognition by $2\%$ for female speakers in the TIMIT dataset, where the majority of training samples are from male voices.

## 1 Introduction

Domain adaptation (Huang et al., 2007; Xue et al., 2008; Ben-David et al., 2010) aims to generalize a model from source domain to a target domain. Typically, the source domain has a large amount of training data, whereas the data are scarce in the target domain. This challenge is typically addressed by learning a mapping between domains, which allows data from the source domain to enrich the available data for training in the target domain. One way of learning such mappings is through Generative Adversarial Networks (GANs Goodfellow et al., 2014) with *cycle-consistency* constraint (CycleGAN Zhu et al., 2017), which enforces that mapping of an example from the source to the target and then back to the source domain would result in the same example (and vice versa for a target example). Due to this constraint, CycleGAN learns to preserve the 'content'[1] from the source domain while only transferring the 'style' to match the distribution of the target domain. This is a powerful constraint, and various works (Yi et al., 2017; Liu et al., 2017; Hoffman et al., 2018) have demonstrated its effectiveness in learning cross domain mappings.

---

[1]Here the content refers to the invariant properties of the data with respect to a task. For example, in image classification the semantic information of an image would be its class. Thus, different task on the same data would result in different semantic information. In this paper we use content and semantic information interchangeably.

Enforcing cycle-consistency is appealing as a technique for preserving semantic information of the data with respect to a task, but implementing it through reconstruction may be too restrictive when data are imbalanced across domains. This is because the reconstruction error encourages exact match of samples from the reverse mapping, which may in turn encourage the forward-mapping to keep the sample close to the original domain. Normally, the adversarial objectives would counter this effect; however, when data from the target domain are scarce, it is very difficult to learn a powerful discriminator that can capture meaningful properties of the target distribution. Therefore, the resulting mappings learned is likely to be sub-optimal. Importantly, for the learned mapping to be meaningful, it is not necessary to have the exact reconstruction. As long as the 'semantic' information is preserved and the 'style' matches the corresponding distribution, it would be a valid mapping.

To address this issue, we propose an augmented cyclic adversarial learning model (ACAL) for domain adaptation. In particular, we replace the reconstruction objective with a task specific model. The model learns to preserve the 'semantic' information from the data samples in a particular domain by minimizing the loss of the mapped samples for the task specific model. On the other hand, the task specific model also serves as an additional source of information for the corresponding domain and hence supplements the discriminator in that domain to facilitate better modeling of the distribution. The task specific model can also be viewed as an implicit way of disentangling the information essential to the task from the 'style' information that relates to the data distribution of different domain. We show that our approach improves the performance by $40\%$ as compared to the baseline on digit domain adaptation. We improve the phoneme error rate by $\sim 5\%$ on TIMIT dataset, when adapting the model trained on one speech from one gender to the other.

## 1.1 RELATED WORK

Our work is broadly related to domain adaptation using neural networks for both supervised and unsupervised domain adaptation.

**Supervised Domain Adaptation** When labels are available in the target domain, a common approach is to utilize the label information in target domain to minimize the discrepancy between source and target domain (Hu et al., 2015; Tzeng et al., 2015; Gebru et al., 2017; Hoffman et al., 2016; Gupta et al., 2016; Ge and Yu, 2017). For example, Hu et al. (2015) applies the marginal Fisher analysis criteria and Maximum Mean Discrepancy (MMD) to minimize the distribution difference between source and target domain. Tzeng et al. (2015) proposed to add a domain classifier that predicts domain label of the inputs, with a domain confusion loss. Gebru et al. (2017) leverages attributes by using attribute and class level classification loss with attribute consistent loss to fine-tune the target model. Our method also employs models from both domains, however, our models are used to assist adversarial learning for better learning of the target domain distribution. In addition, our final model for supervised domain adaptation is obtained by training on data from target domain as well as the transfered data from the source domain, rather than fine-tuning a source/target domain model.

**Unsupervised Domain Adaptation** More recently, various work have taken advantage of the substantial generation capabilities of the GAN framework and applied them to domain adaptation (Liu and Tuzel, 2016; Bousmalis et al., 2017; Yi et al., 2017; Tzeng et al., 2017; Kim et al., 2017; Hoffman et al., 2018). However, most of these works focus on *high-resource unsupervised* domain adaptation, which may be unsuitable for situations where the target domain data are limited. Bousmalis et al. (2017) uses a GAN to adapt data from the source to target domain while simultaneously training a classifier on both the source and adapted data. Our method also employs task specific models; however, we use the models to augment the CycleGAN formulation. We show that having cycles in both directions (*i.e.* from source to target and *vice versa*) is important in the case where the target domain has limited data (see sec. 4). Tzeng et al. (2017) proposes adversarial discriminative domain adaptation (ADDA), where adversarial learning is employed to match the representation learned from the source and target domain. Our method also utilizes pre-trained model from source domain, but we only implicitly match the representation distributions rather than explicitly enforcing representational similarity. Cycle-consistent adversarial domain adaptation (CyCADA Hoffman et al., 2018) is perhaps the most similar work to our own. This approach uses both $\ell_1$ and semantic consistency to enforce cycle-consistency. An important difference in our work is that we also include another cycle that starts from the target domain. This is important because, if the target domain is of low resource, the adaptation from source to target may fail due to the difficulty in learning a good

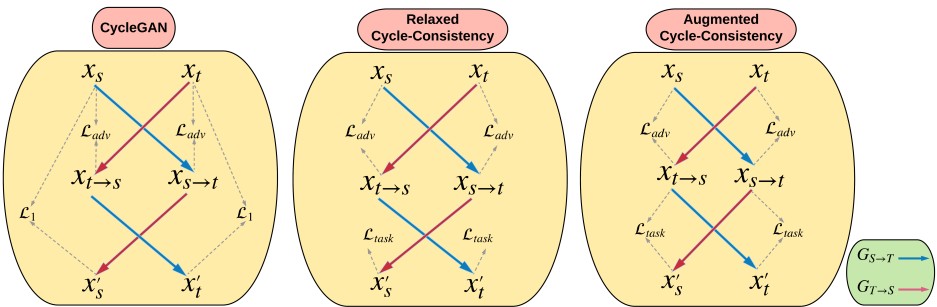

Figure 1: Illustration of proposed approach. Left: CycleGAN (Zhu et al., 2017). Middle: Relaxed cycle-consistent model (RCAL), where the cycle-consistency is enforced through task specific models in corresponding domain. Right: Augmented cycle-consistent model (ACAL). In addition to the relaxed model, the task specific model is also used to augment the discriminator of corresponding domain to facilitate learning. In the diagrams $x$ and $\mathcal{L}$ denote data and losses, respectively. We point out that the ultimate goal of our approach is to use the mapped Source $\rightarrow$ Target samples ($x_{S \mapsto T}$) to augment the limited data of the target domain ($x_T$).

discriminator in the target domain. Almahairi et al. (2018) also suggests to improve CycleGAN by explicitly enforcing content consistency and style adaptation, by augmenting the cyclic adversarial learning to hidden representation of domains.

Our model is different from recent cyclic adversarial learning, due to implicit learning of content and style representation through an auxiliary task, which is more suitable for low resource domains. Using classification to assist GAN training has also been explored previously (Springenberg, 2015; Sricharan et al., 2017; Kumar et al., 2017). Springenberg (2015) proposed CatGAN, where the discriminator is converted to a multi-class classifier. We extend this idea to any task specific model, including speech recognition task, and use this model to preserve task specific information regarding the data.We also propose that the definition of task model can be extended to unsupervised tasks,such as language or speech modeling in domains, meaning augmented unsupervised domain adaptation.

## 2 PRELIMINARIES

### 2.1 GENERATIVE ADVERSARIAL NETWORK

To learn the true data distribution $P_{data}(X)$ in a nonparametric way, Goodfellow et al. (2014) proposed the generative adversarial network (GAN). In this framework, a discriminator network $D(x)$ learns to discriminate between the data produced by a generator network $G(z)$ and the data sampled from the true data distribution $P_{data}(X)$, whereas the generator models the true data distribution by learning to confuse the discriminator. Under certain assumptions (Goodfellow et al., 2014), the generator would learn the true data distribution when the game reaches equilibrium. Training of GAN is in general done by alternately optimizing the following objective for $D$ and $G$.

$$\min_G \max_D V(G, D) = \mathbb{E}_{x \sim P_{data}(X)}\left[\log D(x)\right] + \mathbb{E}_{z \sim P_z(Z)}\left[\log\left(1 - D(G(z))\right)\right] \quad (1)$$

### 2.2 CYCLEGAN

CycleGAN (Zhu et al., 2017) extends this framework to multiple domains, $P_S(X)$ and $P_T(X)$, while learning to map samples back and forth between them. Adversarial learning is applied such that the result mapping from $G_{S \mapsto T}$ will match the target distribution $P_T(X)$, and similarly for the reverse mapping from $G_{T \mapsto S}$. This is accomplished by the following adversarial objectives:

$$\mathcal{L}_{adv}(G_{S \mapsto T}, D_T) = \mathbb{E}_{x \sim P_T(X)}\left[\log D_T(x)\right] + \mathbb{E}_{x \sim P_S(X)}\left[\log\left(1 - D_T(G_{S \mapsto T}(x))\right)\right] \quad (2)$$

$$\mathcal{L}_{adv}(G_{T \mapsto S}, D_S) = \mathbb{E}_{x \sim P_S(X)}\left[\log D_S(x)\right] + \mathbb{E}_{x \sim P_T(X)}\left[\log\left(1 - D_S(G_{T \mapsto S}(x))\right)\right] \quad (3)$$

CycleGAN also introduces cycle-consistency, which enforces that each mapping is able to invert the other. In the original work, this is achieved by including the following reconstruction objective:

$$\mathcal{L}_{cyc}(G_{S \mapsto T}, G_{T \mapsto S}) = \mathbb{E}_{x \sim P_S(X)}[\|G_{T \mapsto S}(G_{S \mapsto T}(x)) - x\|_1]$$
$$+ \mathbb{E}_{x \sim P_T(X)}[\|G_{S \mapsto T}(G_{T \mapsto S}(x)) - x\|_1] \quad (4)$$

Learning the CycleGAN model involves optimizing a weighted combination of the above objectives 2, 3 and 4.

# 3 AUGMENTED CYCLIC ADVERSARIAL LEARNING (ACAL)

Enforcing cycle-consistency using a reconstruction objective (*e.g.* eq. 4) may be too restrictive and potentially results in sub-optimal mapping functions. This is because the learning dynamics of CycleGAN balance the two contrastive forces. The adversarial objective encourages the mapping functions to generate samples that are close to the true distribution. At the same time, the reconstruction objective encourages identity mapping. Balancing these objectives may works well in the case where both domains have a relatively large number of training samples. However, problems may arise in case of domain adaptation, where data within the target domain are relatively sparse.

Let $P_S(X)$ and $P_T(X)$ denote source and target domain distributions, respectively, and samples from $P_T(X)$ are limited. In this case, it will be difficult for the discriminator $D_T$ to model the actual distribution $P_T(X)$. A discriminator model with sufficient capacity will quickly overfit and the resulting $D_T$ will act like delta function on the sample points from $P_T(X)$. Attempts to prevent this by limiting the capacity or using regularization may easily induce over-smoothing and under-fitting such that the probability outputs of $D_T$ are only weakly sensitive to the mapped samples. In both cases, the influence of the reconstruction objective should begin to outweigh that of the adversarial objective, thereby encouraging an identity mapping. More generally, even if we are are able to obtain a reasonable discriminator $D_T$, the support of the distribution learned through it would likely to be small due to limited data. Therefore, the learning signal $G_{S \mapsto T}$ receive from $D_T$ would be limited. To sum up, limited data within $P_T(X)$ would make it less likely that the discriminator will encourage meaningful cross domain mappings.

The root of the above issue in domain adaptation is two fold. First, exact reconstruction is a too strong objective for enforcing cycle-consistency. Second, learning a mapping function to a particular domain which solely depends on the discriminator for that domain is not sufficient. To address these two problems, we propose to 1) use a task specific model to enforce the cycle-consistency constraint, and 2) use the same task specific model in addition to the discriminator to train more meaningful cross domain mappings. In more detail, let $M_S$ and $M_T$ be the task specific models trained on domains $P_S(X, Y)$ and $P_T(X, Y)$, and $\mathcal{L}_{task}$ denotes the task specific loss. Our cycle-consistent objective is then:

$$\mathcal{L}_{RCAL}(G_{S \mapsto T}, G_{T \mapsto S}, M_S, M_T) = \mathbb{E}_{(x,y) \sim P_S(X,Y)} \left[ \mathcal{L}_{task}(M_S(G_{T \mapsto S}(G_{S \mapsto T}(x)), y) \right]$$
$$+ \mathbb{E}_{(x,y) \sim P_T(X,Y)} \left[ \mathcal{L}_{task}(M_T(G_{S \mapsto T}(G_{T \mapsto S}(x)), y) \right] \quad (5)$$

Here, $\mathcal{L}_{task}$ enforces cycle-consistency by requiring that the reverse mappings preserve the semantic information of the original sample. Importantly, this constraint is less strict than when using reconstruction, because now as long as the content matches that of the original sample, the incurred loss will not increase. (Some style consistency is implicitly enforced since each model $M$ is trained on data within a particular domain.) This is a much looser constraint than having consistency in the original data space, and thus we refer to this as the relaxed cycle-consistency objective.

To address the second issue, we augment the adversarial objective with corresponding objective:

$$\mathcal{L}_{ACAL-supervised}(G_{T \mapsto S}, D_S, M_S) = \mathbb{E}_{x \sim P_S(X)} \left[ \log(D_S(x)) \right]$$
$$+ \mathbb{E}_{x \sim P_T(X)} \left[ \log(1 - D_S(G_{T \mapsto S}(x))) \right]$$
$$+ \mathbb{E}_{(x,y) \sim P_S(x,y)} \left[ \mathcal{L}_{task}(M_S(x, y)) \right]$$
$$+ \mathbb{E}_{(x,y) \sim P_T(x,y)} \left[ \mathcal{L}_{task}(M_S(G_{T \mapsto S}(x), y)) \right] \quad (6)$$
$$\mathcal{L}_{ACAL-supervised}(G_{S \mapsto T}, D_T, M_T) = \mathbb{E}_{x \sim P_T(X)} \left[ \log(D_T(x)) \right]$$
$$+ \mathbb{E}_{x \sim P_S(X)} \left[ \log(1 - D_T(G_{S \mapsto T}(x))) \right]$$
$$+ \mathbb{E}_{(x,y) \sim P_T(x,y)} \left[ \mathcal{L}_{task}(M_T(x, y)) \right]$$
$$+ \mathbb{E}_{(x,y) \sim P_S(x,y)} \left[ \mathcal{L}_{task}(M_T(G_{S \mapsto T}(x), y)) \right] \quad (7)$$

Similar to adversarial training, we optimize the above objective by maximizing $D_S$ ($D_T$) and minimizing $G_{T \mapsto S}$ ($G_{S \mapsto T}$) and $M_S(M_T)$. With the new terms, learning of the mapping functions $G$ get assists from both the discriminator and the task specific model. The task specific model learns to capture conditional probability distribution $P_S(Y|X)$ ($P_T(Y|X)$), that also preserves information regarding $P_S(X)$ ($P_T(X)$). This conditional information is different than the information captured through the discriminator $D_S$ ($D_T$). The difference is that the model is only required to preserve

---

**Algorithm 1** Augmented Cyclic Adversarial Learning (ACAL)

---

**Input:** source domain data $P_S(x, y)$, target domain data $P_T(x, y)$, *pretrained* source task model $M_S$
**Output:** target task model $M_T$
**while** *not converged* **do**
    Sample from $(x_s, y_s)$ from $P_S$
    **if** $y_t$ *in* $P_T$ **then**
        *%Supervised%*
        Sample $(x_t, y_t)$ from $P_T$
        Finetune source model $M_S$ on $(x_s, y_s)$ and $(G_{T \mapsto S}(x_t), y_t)$ samples (eq. 6)
        Train task model $M_T$ on $(x_t, y_t)$ and $(G_{S \mapsto T}(x_s), y_s)$ samples (eq. 7)
    **else**
        *%Un-supervised%*
        Sample $x_t$ from $P_T$
        Finetune source model $M_S$ on $(x_s, y_s)$ samples (eq. 8)
        Train task model $M_T$ $(G_{S \mapsto T}(x_s), y_s)$ and $(x_t, M_S(G_{T \mapsto S}(x_t)))$ samples (eq. 9)
    **end**
**end**

---

useful information regarding $X$ respect to predicting $Y$, for modeling the conditional distribution, which makes learning the conditional model a much easier problem. In addition, the conditional model mediates the influence of data that the discriminator does not have access to ($Y$), which should further assist learning of the mapping functions $G_{T \mapsto S}$ ($G_{S \mapsto T}$).

In case of unsupervised domain adaptation, when there is no information of target conditional probability distribution $P_T(Y|X)$, we propose to use source model $M_S$ to estimate $P_T(Y|X)$ through adversarial learning, i.e. $P_T(Y|X) \approx \mathbb{E}_{x \sim P_T(X)}[M_S(G_{S \mapsto T}(x))]$. Therefore, proposed model can be extended to unsupervised domain adaptation, with the corresponding modified objectives:

$$
\begin{aligned}
\mathcal{L}_{ACAL-unsupervised}(G_{T \mapsto S}, D_S, M_S) &= \mathbb{E}_{x \sim P_S(X)}\left[\log(D_S(x))\right] \\
&+ \mathbb{E}_{x \sim P_T(X)}\left[\log(1 - D_S(G_{T \mapsto S}(x)))\right] \\
&+ \mathbb{E}_{(x,y) \sim P_S(x,y)}\left[\mathcal{L}_{task}(M_S(x, y))\right] \quad (8) \\
\mathcal{L}_{ACAL-unsupervised}(G_{S \mapsto T}, D_T, M_T) &= \mathbb{E}_{x \sim P_T(X)}\left[\log(D_T(x))\right] \\
&+ \mathbb{E}_{x \sim P_S(X)}\left[\log(1 - D_T(G_{S \mapsto T}(x)))\right] \\
&+ \mathbb{E}_{(x,y) \sim P_T(x,y)}\left[\mathcal{L}_{task}(M_T(x, M_S(G_{T \mapsto S}(x))))\right] \\
&+ \mathbb{E}_{(x,y) \sim P_S(x,y)}\left[\mathcal{L}_{task}(M_T(G_{S \mapsto T}(x), y))\right] \quad (9)
\end{aligned}
$$

To further extend this approach to semi-supervised domain adaptation, both supervised and unsupervised objectives for labeled and unlabeled target samples are used interchangeably, as explained in Algorithm 1.

## 4 EXPERIMENTS

In this section, we evaluate our proposed model on domain adaptation for visual and speech recognition. We continue the convention of referring to the data domains as 'source' and 'target', where target denotes the domain with either limited or unlabeled training data. Visual domain adaptation is evaluated using the MNIST dataset ($\mathcal{M}$) Lecun et al. (1998), Street View House Numbers (SVHN) datasets ($\mathcal{S}$) Netzer et al. (2011), USPS ($\mathcal{U}$) (Hull, 1994), MNISTM ($\mathcal{MM}$) and Synthetic Digits ($\mathcal{SD}$) (Ganin and Lempitsky, 2014). Adaptation on speech is evaluated on the domain of gender within the TIMIT dataset Garofolo et al. (1993), which contains broadband 16kHz recordings of 6300 utterances (5.4 hours) of phonetically-balanced speech. The male/female ratio of speakers across train/validation/test sets is approximately 70% to 30%. Therefore, we treat male speech as the source domain and female speech as the low resource target domain.

### 4.1 Model Ablations

To get an idea of the contribution from each component of our model, in this section we perform a series of ablations and present the results in Table 1. We perform these ablations by treating SVHN as the source domain and MNIST as the target domain. We down sample the MNIST training data so only 10 samples per class are available during training, denoted as MNIST-(10), which is only 0.17% of full training data. The testing performance is calculated on the full MNIST test set. We use a modified LeNet for all experiments in this ablation. The Modified LeNet consists of two convolutional layers with 20 and 50 channels, followed by a dropout layer and two fully connected layers of 50 and 10 dimensionality.

There are various ways that one may utilize cycle-consistency or adversarial training to do domain adaptation from components of our model. One way is to use adversarial training on the target domain to ensure matching of distribution of adapted data, and use the task specific model to ensure the 'content' of the data from the source domain is preserved. This is the model described in Bousmalis et al. (2017), except their model is originally unsupervised. This model is denoted as $S \to T$ in Table 1. It is also interesting to examine the importance of the double cycle, which is proposed in Zhu et al. (2017) and adopted in our work. Theoretically, one cycle would be sufficient to learn the mapping between domains; therefore, we also investigate the performance of one cycle only models, where one direction would be from source to target and then back, and similarly for the other direction. These models are denoted as (S→T→S)-One Cycle and (T→S→T)-One Cycle in Table 1, respectively. To test the effectiveness of

Table 1: Ablation study results from SVHN (Source) to MNIST (Target). See text for more details. *Note:* The MNIST domain is limited to only 10 samples per class (0.17% of full training dataset), denoted as MNIST-(10). Experiments were performed 4 times with different random sampling for MNIST.

| Domain Adaptation Model | Test Accuracy (%) |
|---|---|
| No Adaptation (trained on SVHN) | 71.11 |
| Target Model (trained on MNIST-(10)) | 79.22±3.98 |
| SVHN+MNIST-(10) | 85.62±1.15 |
| S→T | 69.91±1.56 |
| (S→T→S)-One Cycle | 46.32±2.09 |
| (T→S→T)-One Cycle | 58.34±2.49 |
| (S→T→S)-RCAL (Ours) | **72.51±1.71** |
| (T→S→T)-RCAL (Ours) | 43.56±2.92 |
| (S→T→S)-ACAL (Ours) | **79.40±0.73** |
| (T→S→T)-ACAL (Ours) | 49.81±0.53 |
| CycleGAN | 45.54±1.05 |
| RCAL (Ours) | 88.62±1.77 |
| ACAL (Ours) | **93.90±0.33** |

the relaxed cycle-consistency (eq. 5) and augmented adversarial loss (eq. 6 and 7), we also test one cycle models while progressively adding these two losses. Interestingly, the one cycle relaxed and one cycle augmented models are similar to the model proposed in Hoffman et al. (2018) when their model performs mapping from source to target domain and then back. The difference is that their model is unsupervised and includes more losses at different levels.

As can be seen from Table 1, the simple conditional model performed surprisingly well as compared to more complicated cyclic counterparts. This may be attributed to the reduced complexity, since it only needs to learn one set of mapping. As expected, the single cycle performance is poor when the target domain is of limited data due to inefficient learning of discriminator in the target domain (see section 3). When we change the cycle to the other direction, where there are abundant data in the target domain, the performance improves, but is still worse than the simple one without cycle. This is because the adaptation mapping (*i.e.* $G_{S \mapsto T}$) is only learned via the generated samples from $G_{T \mapsto S}$, which likely deviate from the real examples in practice. This observation also suggests that it would be beneficial to have cycles in both directions when applying the cycle-consistency constraint, since then both mappings can be learned via real examples. The trends get reversed when we are using relaxed implementation of cycle-consistency from the reconstruction error with the task specific losses. This is because now the power of the task specific model is crucial to preserve the content of the data after the reverse mapping. When the source domain dataset is sufficiently large, the cycle-consistency is preserved. As such, the resulting learned mapping functions would preserve meaningful semantics of the data while transferring the styles to the target domain, and *vice versa*. In addition, it is clear that augmenting the discriminator with task specific loss is helpful for learning adaptations. Furthermore, the information added from the task specific model is clearly beneficial for improving the adaptation performance, without this none of the models outperform the baseline

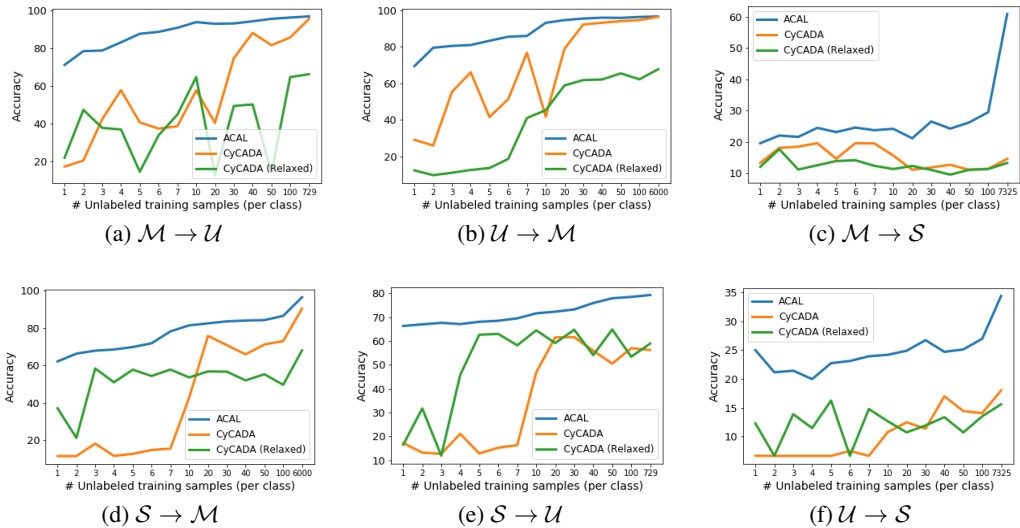

Figure 2: Comparison of adaptation robustness between CyCADA (Hoffman et al., 2018), CyCADA with no $\ell_1$ reconstruction loss (Relaxed), and ACAL algorithms for variable number of unsupervised target samples. *Note: No labeled sample is used.*

model, where no adaptation is performed. Last but not least, it is also clear from the results that using task specific model improves the overall adaptation performance.

To further evaluate the effectiveness of using task-specific loss with two cycles for low-resource unsupervised domain adaptation scenario, we comapre our model with CyCADA (Hoffman et al., 2018), and when no reconstruction loss is used in CyCADA, referred as "CyCADA (Relaxed)". The latter resembles the $(S \to T \to S)$-ACAL in Table 1, but with a different semantic loss. As shown in Figure 2, CyCADA model and its relaxed variation fail to learn a good adaptation, where target domain contains few unlabaled samples per class. Additionally, CyCADA models show high instability in low-resource situation. As described in section 1.1, instability is an expected behvaiour of CyCADA when having limited target data, because the source to target cycle fails to preserve consistency, due to weak target domain discriminator. However, ACAL model indicates stable and consistent performance, due to proper use of source classifier to enforce consistency, rather than relying on target and source discriminators.

## 4.2 VISUAL DOMAIN ADAPTATION

In this section, we experiment on domain adaptation for the task of digit recognition. In each experiment, we select one domain (MNIST, USPS, MNISTM, SVHN, Synthetic Digits) to be the target. We conduct three types of domain adaptation, i.e. *low-resource supervised*, *high-resource unsupervised*, and *low-resource semi-supervised* adaptation. The evaluation results are based on not using any data augmentation.

**Low-resource supervised adaptation:** In this setting, we sub-sample the target to contain only a few labeled samples per class, and using the other full dataset as the source domain. In this setting, no unlabeled sample is used. Comparison with recent low resource domain adaptation, FADA (Motiian et al., 2017) for MNIST, USPS, and SVHN adaptation is shown in Figure 3. To provide more baselines, we also compared with model trained only on limited target data, and on combination of both labeled source and limited target domains. As shown in Figure 4, ACAL outperforms FADA and two other baselines in all adaptations.

**High-resource unsupervised adaptation;** Here, we use the whole target domain with no label. Evaluation results on all adaptation directions are presented in Table 2 and Table 7 (Appendix A). It is evident that ACAL model performance is on par with the state of the art unsupervised approaches, and outperforms on MNIST→USPS and Syn-Digits→SVHN. It is worth mentioning that Shu et al.

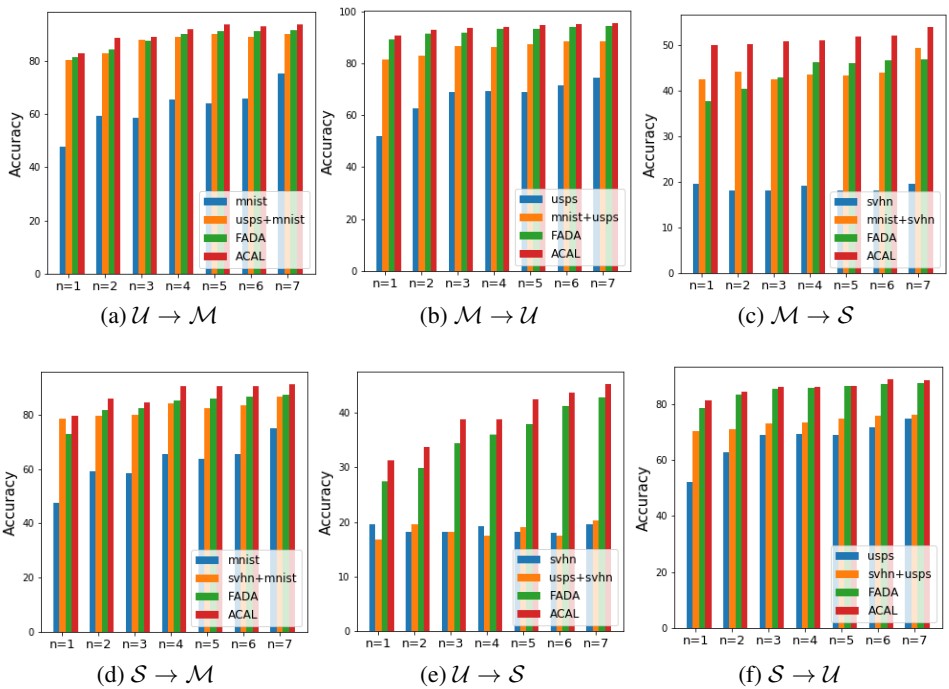

Figure 3: Low-resource supervised Domain Adaptation on MNIST ($\mathcal{M}$), USPS ($\mathcal{U}$) and SVHN ($\mathcal{S}$) datasets. FADA model refers to Motiian et al. (2017). $n = 1$ means 1 labeled example per class. No unlabeled target sample is used.

(2018) improved their VADA adversarial model using natural gradient as teacher-student training, which is not directly comparable to adversarial approaches. Moreover, the *source-only* baseline of (Shu et al., 2018) is stronger than the reported unsupervised approaches, as well as our baseline.

**Low-resource semi-supervised adaptation:** We also evaluate the performance of ACAL algorithm when there are limited labeled and unlabeled target samples in Table 6 (Appendix A). In case of MNIST→USPS, our model outperforms many high-resource unsupervised domain adaptation in Table 2 by using $< 1000$ unlabeled samples only.

### 4.3 SPEECH DOMAIN ADAPTATION

We also apply our proposed model to domain adaptation in speech recognition. We use TIMIT dataset, where the male to female speaker ratio is about $7 : 3$ and thus we choose the data subset from male speakers as the source and the subset from female speakers as the target domain. We evaluate performance on the standard TIMIT test set and use phoneme error rate (PER) as the evaluation metric. Spectrogram representation of audio is chosen for model evaluation. As demonstrated by Hosseini-Asl et al. (2018), multi-discriminator training significantly impacts adaptation performance. Therefore, we used the multi-discriminator architecture as the discriminator for the adversarial loss in our evaluation. Our task-specific model is a pre-trained speech recognition model within each domain in this set of experiments.

The result are shown in Table 3. We observe significant performance improvements over the baseline model as well as comparable or better performance as compared to previous methods. It is interesting to note that the performance of the proposed model on the adapted male ($\mathcal{M} \rightarrow \mathcal{F}$) almost matches the baseline model performance, where the model is trained on true female speech. In addition, the performance gap in this case is significant as compared to other methods, which suggests the adapted distribution is indeed close to the true target distribution. In addition, when combined with more data, our model further outperforms the baseline by a noticeable margin.

Table 2: *High-resource unsupervised* domain adaptation between MNIST ($\mathcal{M}$), USPS ($\mathcal{U}$), MNIST-M ($\mathcal{MM}$), SVHN ($\mathcal{S}$), Synthetic Digits ($\mathcal{SD}$). Note: Direction indicates source→target adaptation direction. VADA (Shu et al., 2018) used a stronger *source-only* baseline on $\mathcal{S} \rightarrow \mathcal{M}$ (82.4 accuracy) compared to other approaches. *Note:* No data augmentation is used in our experiments.

| Model | Direction | Domain pairs | | | |
| --- | --- | --- | --- | --- | --- |
| | | $\mathcal{M} - \mathcal{U}$ | $\mathcal{M} - \mathcal{MM}$ | $\mathcal{M} - \mathcal{S}$ | $\mathcal{S} - \mathcal{SD}$ |
| Source-only | → | 83.46 | 59.55 | 38.03 | 90.32 |
| | ← | 71.14 | 98.36 | 71.11 | 88.17 |
| DA (Häusser et al., 2017) | → | - | 89.53 | - | - |
| | ← | - | - | **97.6** | 91.86 |
| VADA (Shu et al., 2018) | → | - | 95.7 | **73.3** | - |
| | ← | - | - | 94.5 | 94.9 |
| Self-ensembling (MT+CT) (French et al., 2018) | → | 88.14 | - | 33.87 | - |
| | ← | 92.35 | - | 93.33 | 96.01 |
| DupGAN (Hu et al., 2018) | → | 96.01 | - | 62.65 | - |
| | ← | **98.75** | - | 92.46 | - |
| CyCADA (Hoffman et al., 2018) | → | 95.6 | 57.21 | 14.56 | 81.19 |
| | ← | 96.5 | 94.57 | 90.4 | 72.94 |
| SBADA-GAN (Russo et al., 2018) | → | 97.6 | **99.4** | 61.1 | - |
| | ← | 95.0 | - | 76.1 | - |
| ACAL (Ours) | → | **98.31** | 97.29 | 60.85 | **96.43** |
| | ← | 97.16 | **99.26** | 96.51 | **97.98** |
| Target-only (completely supervised) | → | 96.26 | 98.19 | 93.38 | 98.60 |
| | ← | 99.49 | 99.49 | 99.49 | 93.38 |

Table 3: Speech domain adaptation results on TIMIT. We treat Male ($\mathcal{M}$) and Female ($\mathcal{F}$) voices for the source and target domains, respectively, based on the intrinsic imbalance of speaker genders in the dataset (about 7 : 3 male/female ratio). For the evaluation metric, lower is better.

| Training Set | Domain Adaptation Model | Female (PER) | |
| --- | --- | --- | --- |
| | | Val | Test |
| $\mathcal{M}$ | - | 35.70 | 30.69 |
| $\mathcal{F}$ (Baseline model) | - | 24.51 | 23.22 |
| | CycleGAN (Zhu et al., 2017) | 32.95 | 30.07 |
| $\mathcal{M} \rightarrow \mathcal{F}$ | FHVAE (Hsu et al., 2017) | – | 26.2 |
| | MD-CycleGAN (Hosseini-Asl et al., 2018) | 28.80 | 25.45 |
| | ACAL (Ours) | **24.86** | **23.46** |
| | CycleGAN (Zhu et al., 2017) | 28.32 | 28.43 |
| $\mathcal{F} + (\mathcal{M} \rightarrow \mathcal{F})$ | MD-CycleGAN (Hosseini-Asl et al., 2018) | 21.15 | 19.08 |
| | ACAL (Ours) | **20.32** | **19.02** |
| $\mathcal{F} + \mathcal{M}$ | - | 20.63 | 20.52 |
| | CycleGAN (Zhu et al., 2017) | 21.03 | 22.81 |
| $\mathcal{F} + \mathcal{M} + (\mathcal{M} \rightarrow \mathcal{F})$ | MD-CycleGAN (Hosseini-Asl et al., 2018) | 20.26 | 19.60 |
| | ACAL (Ours) | **20.02** | **18.44** |

## 5 CONCLUSION AND FUTURE WORK

In this paper, we propose to use augmented cycle-consistency adversarial learning for domain adaptation and introduce a task specific model to facilitate learning domain related mappings. We enforce cycle-consistency using a task specific loss instead of the conventional reconstruction objective. Additionally, we use the task specific model as an additional source of information for the discriminator in the corresponding domain. We demonstrate the effectiveness of our proposed approach by evaluating on two domain adaptation tasks, and in both cases we achieve significant performance improvement as compared to the baseline.

By extending the definition of task-specific model to unsupervised learning, such as reconstruction loss using autoencoder, or self-supervision, our proposed method would work on all settings of domain adaptation. Such unsupervised task can be speech modeling using wavenet (van den Oord et al., 2016), or language modeling using recurrent or transformer networks (Radford et al., 2018).

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

## APPENDIX A   DIGIT DOMAIN ADAPTATION ANALYSIS

In this section, we evaluate domain adaptation for MNIST↔SVHN for comparison with CycleGAN, as well as the relaxed version of the cycle-consistent objective (Relaxed-Cyc, see eq. 5 in section 3). For the former, $\ell_1$ reconstruction loss is replaced with the model loss in order to encouraging cycle-consistency. We also experiment with two different task specific models $M$: specifically, DenseNet (Huang et al., 2017, representing a relatively complex architecture) and a modified LeNet (representing a relatively simple architecture, see section 4.1).

Table 4 and  5 show the results on augmenting the low resource MNIST and SVHN with the complementary high resource domain. This approach improves test performance of the target classifier by a large margin, compared to when trained only using the target domain data. We observe that training a more complicated deep model for the target domain weakens this effect. As shown in Table 4, using DenseNet as a classifier on MNIST (target) achieves $\approx 24\%$ lower test classification accuracy than using a variant of LeNet. This difference likely reflects differences in the two architectures' degree of overfitting. Overfitting will produce a false gradient signal during cycle adversarial learning (when classifying the adapted source examples). Based on this observation, we use a comparatively simpler LeNet architecture with SVHN as the target domain (see Table 5). Using our proposed approach, SVHN test performance improves by $27\%$ over domain adaptation using CycleGAN. We also include some qualitative results when performing domain adaptation from SVHN (source) to MNIST (target), as shown in Figure 5. We also compare the performance with different number of labeled target samples in Figure 4. It indicates the improvement on generalization performance of target model using Augmented cyclic adaptation, with variable labeled target domain on MNIST and SVHN datasets. Evaluation of semi supervised adaptation is presented in Table 6.

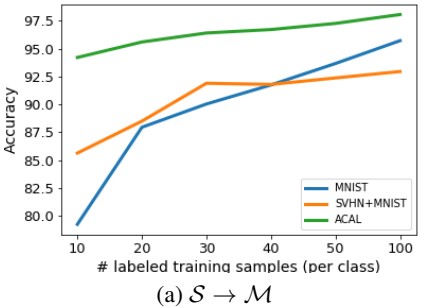 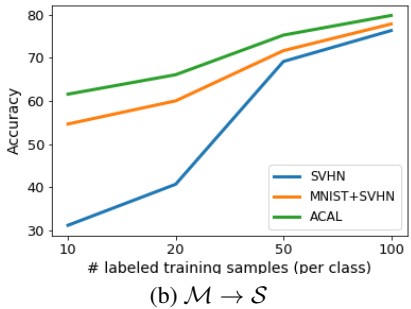

(a) $\mathcal{S} \rightarrow \mathcal{M}$ (b) $\mathcal{M} \rightarrow \mathcal{S}$

Figure 4: Performance comparison of proposed ACAL algorithm on SVHN ($\mathcal{S}$) and MNIST ($\mathcal{M}$) with baselines using different numbers of labeled training sample (per class) in target domain for (a) $S \rightarrow M$ and (b) $M \rightarrow S$ adaptation. (Best viewed in color)

Table 4: Visual domain adaptation results from SVHN to MNIST (Low resource). No adaptation denotes model trained on the source domain (SVHN) and target model refers to model trained on the target domain (MNIST). *Note:* MNIST (Low resource) domain contains only 10 labeled sampels per class (MNIST-(10)), the experiments was performed 4 times with different random sampling for MNIST.

| | MNIST Test (%) | |
| Domain Adaptation Model | LeNet (Modified) | DenseNet |
| --- | --- | --- |
| No Adaptation (trained on SVHN) | 71.11 | 56.92 |
| Target Model (trained on MNIST-(10)) | 79.22±3.98 | 39.89±0.84 |
| CycleGAN | 45.54±1.05 | 28.52±1.65 |
| RCAL (Ours) | 84.62±1.77 | 44.36±3.42 |
| ACAL (Ours) | **93.90±0.33** | **69.47±4.66** |

Table 5: Visual domain adaptation results from MNIST to SVHN (Low resource). No adaptation denotes model trained on the source domain (MNIST) and target model refers to model trained on the target domain (SVHN). *Note:* SVHN (Low resource) domain contains only 50 images per class (SVHN-(50)), the experiments was performed 4 times with different random sampling for SVHN.

| Domain Adaptation Model | SVHN Test (%) LeNet (modified) |
|---|---|
| No Adaptation (trained on MNIST) | 38.03 |
| Target Model (trained on SVHN-(50)) | 70.20±2.35 |
| CycleGAN | 66.75±2.02 |
| RCAL (Ours) | 72.13±0.91 |
| ACAL (Ours) | **74.61±0.43** |

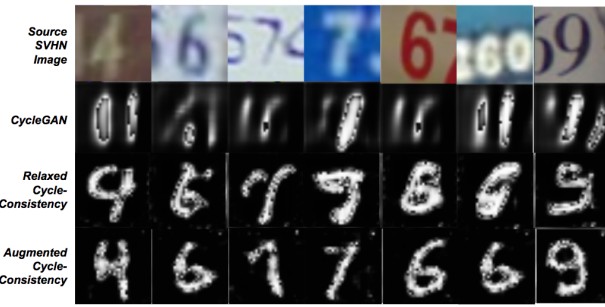

Figure 5: Qualitative comparison of domain adaptation for experimental models. Each column illustrates the mapping performed by each of the models from the original SVHN image (source domain) to MNIST (target domain, 10 labeled samples per class in total). It can be seen that the augmented cycle-consistent model is able to preserve most of the semantic information, while still approximately match the target distribution.

Table 6: *Low-resource semi and unsupervised* domain adaptation on MNIST ($\mathcal{M}$), USPS ($\mathcal{U}$) and SVHN ($\mathcal{S}$) datasets. *Note:* $n = 10$ means 10 samples per class, and $10\%$ denotes the percentage of target samples (per class) which have labels. $0\%$ corresponds to low-resource unsupervised adaptation.

| # target samples per class | $\mathcal{S} \to \mathcal{M}$ 0% | 10% | $\mathcal{M} \to \mathcal{U}$ 0% | 10% | $\mathcal{U} \to \mathcal{M}$ 0% | 10% | $\mathcal{S} \to \mathcal{U}$ 0% | 10% |
|---|---|---|---|---|---|---|---|---|
| $n = 10$ | 81.43 | 77.63 | 93.86 | 94.01 | 93.22 | 94.89 | 71.54 | 75.98 |
| $n = 50$ | 84.26 | 87.22 | 95.61 | 94.17 | 95.93 | 96.83 | 77.87 | 86.19 |
| $n = 100$ | 86.49 | 91.75 | 96.31 | 96.01 | 96.43 | 96.92 | 78.42 | 89.03 |
| $n = $ full train | 96.51 | 99.41 | 96.91 | 95.71 | 96.74 | 98.45 | 79.23 | 93.17 |

Table 7: *High-resource unsupervised* domain adaptation between MNIST ($\mathcal{M}$), USPS ($\mathcal{U}$), MNIST-M ($\mathcal{MM}$), SVHN ($\mathcal{S}$), Synthetic Digits ($\mathcal{SD}$). Note: Direction indicates source→target adaptation direction.

| Model | Direction | $\mathcal{M} - \mathcal{SD}$ | $\mathcal{U} - \mathcal{MM}$ | $\mathcal{U} - \mathcal{S}$ | $\mathcal{U} - \mathcal{SD}$ | $\mathcal{MM} - \mathcal{S}$ | $\mathcal{MM} - \mathcal{SD}$ |
|---|---|---|---|---|---|---|---|
| Source-only | → | 49.71 | 36.69 | 25.11 | 34.90 | 42.82 | 57.53 |
| | ← | 84.44 | 79.22 | 63.73 | 80.72 | 53.21 | 65.38 |
| ACAL (Ours) | → | 68.90 | 63.65 | 34.35 | 42.95 | 65.94 | 65.08 |
| | ← | 92.34 | 94.81 | 79.23 | 91.88 | 69.47 | 78.71 |
| Target-only | → | 98.60 | 98.19 | 93.38 | 98.60 | 93.38 | 98.60 |
| | ← | 99.49 | 96.26 | 96.26 | 96.26 | 98.19 | 98.19 |

## APPENDIX B    SPEECH DOMAIN MODELS IMPLEMENTATION

In this section, the detail of CycleGAN and speech model architectures are explained. The size of the convolution layer are denoted by the tuple (C, F, T, SF, ST), where C, F, T, SF, and ST denote number of channels, filter size in frequency dimension, filter size in time dimension, stride in frequency dimension and stride in time dimension respectively. Architecture of CycleGAN model is based on Zhu et al. (2017) with modifications mentioned in Hosseini-Asl et al. (2018). Both generators in CycleGAN are based on U-net Ronneberger et al. (2015) architecture with 4 layers of convolution of sizes (8,3,3,1,1), (16,3,3,1,1), (32,3,3,2,2), (64,3,3,2,2), followed by corresponding deconvolution layers. To increase stability of adversarial training, as proposed by Hosseini-Asl et al. (2018), the discriminator output is modified to predict a single scalar as real/fake probability. Discriminator has 4 convolution layers of sizes (8,4,4,2,2), (16,4,4,2,2), (32,4,4,2,2), (64,4,4,2,2), as default kernel and stride sizes in Hosseini-Asl et al. (2018). ASR model is implemented based on Zhou et al. (2017), which is trained only with maximum likelihood. The model includes one convolutional layer of size (32,41,11,2,2), and five residual convolution blocks of size (32,7,3,1,1), (32,5,3,1,1), (32,3,3,1,1), (64,3,3,2,1), (64,3,3,1,1) respectively. Convolutional layers are followed by 4 layers of bidirectional GRU RNNs with 1024 hidden units per direction per layer. Finally, a fully-connected hidden layer of size 1024 is used as the output layer.

### B.1    QUALITATIVE EVALUATION OF DOMAIN ADAPTATION

In this section we show some qualitative results on transcriptions produced from different models.

Table 8: ASR prediction improvement on low resource Female domain (TIMIT), when augmented with adapted audios from high resource Male domain

| | | Train on Female + (Male→Female) |
|---|---|---|
| Test on Female | True | sil dh ah m aa r n ih ng sil d uw aa n dh ah s sil p ay dx er w eh sil g l ih s eh n sil d ih n dh ah s ah n sil |
| | No adaptation | sil dh ah m aa r n ih ng sil d uw aa m ih s sil b ay er w ih sil b z l ih s ih n d ih n s ah n sil |
| | CycleGAN | sil dh ih m aa r n ih ng sil d ih ah n dh ih s sil p ay ih w r eh sil dh l dh ih s ih n sil d ih n s ay n sil |
| | ACAL | sil dh ah m aa r n ih ng sil d uw ah n dh ih s sil b ay dx y er w eh sil b l ih s ih n sil d ih n ih s ah n sil |
| | True | sil iy v ih n ah s ih m sil p l v ah sil k ae sil b y ih l eh r iy sil k ah n sil t ey n sil t s ih m sil b l z sil |
| | No Adaptation | sil iy dh ih n ah s ih m v l v ow sil k ae sil b y ih l eh r iy sil k eh n sil t ey n s ih m sil b l z sil |
| | CycleGAN | sil iy ih m ah s eh m sil p l v dh aa sil k ey sil b y ih r ey ey sil k ih n sil t r ey n sil s ih m sil b ah l z sil |
| | ACAL | sil iy v ih n ah s ih m sil p l v ow sil k ae sil b y ih l eh r iy sil k ih n sil t ay ey n s ih m sil b l z sil |
| | True | sil dh ah f aa sil p r ih v ih n ih sil dh ih m f r ah m er r aa v ih ng aa n sil t aa m sil |
| | No Adaptation | sil dh ah f aa sil p er z ih n ih n sil dh ih m z er v er r aa v iy ng aa n sil t ay m sil |
| | CycleGAN | sil b er f aa sil p r ih th iy n m ih sil b ih ih m n sil f r eh m er r aw n iy ng er n sil t er m sil |
| | ACAL | sil dh ih f aa l sil p r ih z ih n ih sil dh iy ih m f er m er r aa dh ih ng aa n sil t ah m sil |
| | True | sil ch iy sil s sil t aa sil k ih ng z r ah n dh ih f er s sil t ay m dh eh r w aa r n sil |
| | No Adaptation | sil ch iy sil ch s sil t aa sil k ih n ng z r ah m dh ah f er s sil t aa m dh eh w ah r n sil |
| | CycleGAN | sil ch iy sil ch s sil t aa sil k ih ng z r ah n dh ih f er ih s sil t ay n dh eh r w aa r ng sil |
| | ACAL | sil sh iy sil ch s sil t aa sil k ih ng z r ah m dh ah f er s sil t ay m dh eh r w aa r n sil |
| | True | sil d ow n sil d uw sil ch aa r l iy z sil d er dx iy sil d ih sh ih z sil |
| | No Adaptation | sil d ow sil d uw sil ch er l iy s sil t er dx iy sil d ey sh ih z sil |
| | CycleGAN | sil dh aw sil d ih sil ch aa r l iy s sil t er dx iy sil d ih sh iy z sil |
| | ACAL | sil d ow n sil d uw sil ch er l iy s sil t er dx iy sil d eh sh ih z sil |
| | True | sil k ae l s iy ih m ey sil s sil b ow n z n sil t iy th s sil t r aa ng sil |
| | No Adaptation | sil k eh l s iy ih m ey sil k s sil b ow n z ih n sil t iy sil s sil t r aa l sil |
| | CycleGAN | sil t aw s iy ih m n m ey sil k s sil b ow n z ih n sil t iy sil s sil t r aa ng sil |
| | ACAL | sil k aw s iy ih m ey sil k s sil b ow n z ih n sil t iy sil s sil t r aa ng sil |

