# OpenReview forum: "Augmented Cyclic Adversarial Learning for Low Resource Domain Adaptation"
_ICLR.cc/2019/Conference_

### Official Review · AnonReviewer3 · 2018-10-30
**Well-motivated approach, but limited novelty and experiments**

**Rating:** 5
**Confidence:** 4

**Review:**

This paper introduces a domain adaptation approach based on the idea of Cyclic GAN. Two different algorithms are proposed. The first one incorporates a semantic consistency loss based on domain-specific classifiers acting on full cycles of the of the generators. The second one also makes use of domain-specific classifiers, but acting either directly on the training samples or on the data mapped from one domain to the other.

Strengths:
- The different terms in the proposed loss functions are well justified.
- The results on low-resources supervised domain adaptation indicate that the method works better than the that of Motiian et al. 2017.

Weaknesses:
- Novelty is limited: The two algorithms are essentially small modification of the semantic consistency term used in Hoffman et al. 2018. They involve making use of both the source and target classifiers, instead of only the source one, and, for the relaxed version, making use of complete cycles instead of just one mapping from one domain to the other. While the modifications are justified, I find this a bit weak for ICLR.

- It is not clear to me why it is worth presenting the relaxed cycle-consistency object, since it always yields worse results than the augmented one. In fact, at first, I though both objectives would be combined in a single loss, and was thus surprised not to see Eq. 5 appear in Algorithm 1. It only became clear when reading the experiments that the authors were treating the two objectives as two different algorithms. Note that, in addition to not performing as well as the augmented version, it is also unclear how the relaxed one could work in the unsupervised scenario.

- Experiments:
* In 4.1, the authors mention that 10 samples per class are available in the target domain. Are they labeled or unlabeled? If labeled, are additional unlabeled samples also used?
* In Table 1, and in Table 3, is there a method that corresponds to CyCADA? I feel that this comparison would be useful considering the similarity. That said, I also understand that CyCADA uses both a reconstruction term (as in Eq. 4) and a semantic consistency one, whereas here only a semantic reconstruction term is used. I therefore suggest the authors to also compare with a baseline that replaces their objective with the semantic consistency one of CyCADA, i.e., CyCADA without reconstruction term.
* In 4.2, it is again not entirely clear if the authors use only the few labeled samples, or if this is complemented with additional unlabeled samples. In any event, does this reproduce the setting used by Motiian et al. 2017?
* As the argument is that the proposed loss is better than the reconstruction one and that of Hoffman et al. 2018 for low-resource supervised adaptation, it would be worth demonstrating this empirically in Table 2.

Summary:
The proposed objective functions are well motivated, but I feel that novelty is too limited and the current set of experiments not sufficient to warrant publication at ICLR.

After Response:
After the authors' response/discussion, while I appreciate the additional results provided by the authors, I still feel that the contribution is a bit weak for ICLR.

---

> ### Author Response · Authors · 2018-11-18
> **Response to AnonReviewer3 comments and suggestions (1/2)**
>
> Comment on Weakness, and similarity to CyCADA model:
> To differentiate between our model and CyCADA, below is the detail of two models and how they perform semantic consistency, and enforcing style adaptation
> CyCADA:
> semantic (content) consistency is enforced by two loss; reconstruction loss (CycleGAN); and additionally using reconstruction at feature level.
> Style adaptation is enforced using adversarial learning on pixel (observation) and feature (hidden) space. Therefore, it need to learn additional model for representing data in feature space.
>
> Augmented-Cyc:
> semantic consistency is shown to be achieved by only using auxiliary task loss for each cycle.
> Style adaptation is achieved by using adversarial learning on pixel (observation) space only.
> We use cycles in both direction to achieve robust performance in low resource (either supervised or unsupervised) setting.
>
> Therefor, CyCADA requires an additional adversarial learning at feature space, while our model achieve this by only adaptation at observation space. Moreover, to compare the performance of the two model on variable-size target domain, we added more experiments for low resource unsupervised adaptation (see Figure 2). It is evident that CyCADA model fails to provide suitable adaptation, while our model outperforms by large margin, when target domain data is small
> Note: Both our ablation (see Table 1) and additional experiments (see Figure 2) suggest the benefit of using two cycles for low resource situation, whether supervised or unsupervised. Therefore, we think this is an important aspect for robust domain adaptation under resource constraint.
>
> Comment on relaxed cycle consistency:
> The main purpose of presenting relaxed-consistency results in ablation study is to demonstrate the effectiveness of using auxiliary task loss in any or both cycles, rather than L1 reconstruction loss. We have only evaluated relaxed-consistency in low-resource supervised setting, and it is not evaluated for unsupervised adaptation. In unsupervised setting, we are using source classifier M_{S} as pseudo-labeler of target samples.
>
> Note: In this setting, if we turn off using task model M_{T} to be trained using source data, this is similar to using relaxed version in unsupervised adaptation
>
>
> Comments on Experiments:
>
> For all low resource target domain experiment, only the denoted number of samples are used, irrespective if they are labeled or not. For example, in supervised case, 10 labeled sample per class means we only use 10 labeled samples per class in the target domain is used and no other data is used in the target domain. Similarly for unsupervised case, 5 samples per class means only used 5 unsupervised samples from target domain.
>
> - Section 4.1:  we only used 10 labeled sample per class. In this experiment, NO unlabeled data is used.
>
> - Table 1: this table is intended for ablation of our model.
>
> - Table 3: we have added CyCADA results in this table for comparison. To directly compare our model with CyCADA, we added new experiments on variable-size target domain which is presented in Figure 2.
>
> - Section 4.2: Table 2 is replaced with Figure 3, for low-resource supervised adaptation. In this experiment, no unlabeled data is used, and it is a direct comparison between our model and FADA (Motiian et al. 2017)
>
> In Figure 2, we have shown the benefit of the proposed auxiliary task-specific loss to reconstruction loss (CyCADA) on low-resource unsupervised domain adaptation.

---

> > ### Author Response · Authors · 2018-11-25
> > **Response to AnonReviewer3 comments and suggestions  (2/2)**
> >
> > Comment on Experiments:
> >
> > Table 1: we also added the results for CyCADA model with no reconstruction loss in Figure 2, referred to as "CyCADA (Relaxed)", to provide more baselines.

---

> > > ### Comment · AnonReviewer3 · 2018-11-28
> > > **Response**
> > >
> > > I read the authors' response and have a couple more comments:
> > >
> > > - The thing that bothered me regarding novelty, and that the authors did not comment on in their response, is that CyCADA also uses a semantic consistency loss. This is the loss in Eq. 4 of the CyCADA paper, which looks very similar to the last two terms in Eqs. 6 and 7 of this submission. I understand that there are differences, but I find them a bit thin as ICLR contributions.
> > >
> > > - I appreciate the comparison to CyCADA in Fig. 2 and Table 2. The results in Fig. 2, however, only represent a subset of the pairs considered in Fig. 3. I would suggest including all the pairs. Considering that the authors were able to compute CyCADA results for Fig. 2, I imagine that it would also be possible for them to fill the missing CyCADA values in Table 2.

---

> > > > ### Author Response · Authors · 2018-12-01
> > > > **Comment on Novelty, similarity to CyCADA equations**
> > > >
> > > > We would like emphasize again that our focus in this paper is for low resource domain adaptation, and our main contribution and novelty is in the introduction of two cycles for low resource domain adaptation. From our experiments, it is clear that the introduction of an additional cycle is necessary to get robust performance in low resource settings, irrespective of whether learning is supervised or unsupervised. The two cycle structure is not present in CyCADA. While this may seem like a subtle difference, the benefit of the additional cycle is clear, as shown in both our ablation study (Table 1) and the comparison between CyCADA and our method (Fig 2). Our intuition for this benefit is that conversion from both direction makes more training data available for the model, which results in more robust models. This improvement is both consistent and significant as compared to CycleGAN and other methods in various experiments under different settings, which shows the significance of the change.
> > > >
> > > > We agree that the last two loss terms are similar from ours eq. 6 and 7 to eq. 4 in CyCADA, because the motivation of these terms are similar. However, there are still some differences:
> > > > In CyCADA paper they would like the classifier to perform consistently across domains, whereas ours tries to make sure that the generator between domains preserves task specific information. And the consistency is preserved through two task losses. Ours uses the true label when available, whereas CyCADA uses model output.
> > > > CyCADA’s model does not get tuned in the consistency loss, whereas in our methods all models are tuned.
> > > >
> > > > Additionally, CyCADA has another discriminator at the feature level, which helps features transfer between domains, and we only have one in the data space. Our design is simpler and more robust in this case, since it is non-trivial to design a good discriminator that works well when one departs from static data like images--for example, when modeling sequential data such as text or audio.

---

> > > > > ### Comment · AnonReviewer3 · 2018-12-02
> > > > > **One more question**
> > > > >
> > > > > Thanks for the new results.
> > > > >
> > > > > Above, you stated that:
> > > > > "CyCADA’s model does not get tuned in the consistency loss, whereas in our methods all models are tuned."
> > > > >
> > > > > Can you expand on this? It is not clear to me what you mean.

---

> > > > > > ### Author Response · Authors · 2018-12-03
> > > > > > **Reply**
> > > > > >
> > > > > > In section 3 of CyCADA's paper, under equation 3, "...We pretrain a source task model f_S, fixing the weights, we use this model as a noisy labeler ...". So for eq. 4 (i.e. semantic loss) in their work, only G_t_s and G_s_t is optimized.

---

> > > > ### Author Response · Authors · 2018-12-01
> > > > **Comment on additional experiments on Fig 2 and Table 2**
> > > >
> > > > We have ran additional experiments and the updated Fig. 2 and Table 2 is provided in the following links.
> > > > Figure 2: https://bit.ly/2E812qf
> > > > Table 2:  https://bit.ly/2QwEogQ

---

### Official Review · AnonReviewer2 · 2018-11-02

**Rating:** 6
**Confidence:** 4

**Review:**

The authors propose an extension of cycle-consistent adversarial adaptation methods in order to tackle domain adaptation in settings where a limited amount of supervised target data is available (though they also validate their model in the standard unsupervised setting as well). The method appears to be a natural generalization/extension of CycleGAN/CyCADA. It uses the ideas of the semantic consistency loss and training on adapted data from CyCADA, but "fills out" the model by applying these techniques in both directions (whereas CyCADA only applied them in the source-to-target direction).

The writing in this paper is a little awkward at times (many omitted articles such as "the" or "a'), but, with a few exceptions, it is generally easy to understand what the authors are saying. They provide experiments in a variety of settings in order to validate their model, including both visual domain adaptation and speech domain adaptation. The experiments show that their model is effective both in low-resource supervised adaptation settings as well as high-resource unsupervised adaptation settings. An ablation study, provided in Section 4.1, helps to understand how well the various instantiations of the authors' model perform, indicating that enforcing consistency in both methods is crucial to achieving performance beyond the simple baselines.

It's a little hard to understand how this method stands in comparison to existing work. Table 3 helps to show that the model can scale up to the high-resource setting, but it would also be nice to see the reverse: comparisons against existing work run in the limited data setting, to better understand how much limited data negatively impacts the performance of models that weren't designed with this setting in mind.

I would've also liked to see more comparisons against the simple baseline of a classifier trained exclusively on the available supervised target data, or with the source and target data together—in my experience, these baselines can prove to be surprisingly strong, and would give a better sense of how effective this paper's contributions are. This corresponds to rows 2 and 3 of Table 1, and inspection of the numbers in that table shows that the baseline performance is quite strong even relative to the proposed method, so it would be nice to see these numbers in Table 2 as well, since that table is intended to demonstrate the model's effectiveness across a variety of different domain shifts.

While it's nice that the model is experimentally validated on the speech domain, the experiment itself is not explained well. The speech experiments are hard to understand—it's unclear what the various training sets are, such as "Adapted Male" or "All Data," making it hard to understand exactly what numbers should be compared. Why is there no CycleGAN result for "Female + Adapted Male," or "All Data + Adapted Male," for example? The paper would greatly benefit from a more careful explanation and analysis of this experimental setting.

Ultimately, I think the idea is a nice generalization of previous work, and the experiments seem to indicate that the model is effective, but the limited scope of the experiments prevent me from being entirely convinced. The inclusion of additional baselines and a great deal of clarification on the speech experiments would improve the quality of this paper enormously.

---

Update: After looking over the additional revisions and experiments, I'm bumping this to a weak accept. I agree with reviewer 3 that novelty is not the greatest, but there is a useful contribution here, and the demonstration of its effectiveness on low resource settings is valuable, since in a practical setting it is usually feasible to manually label a few examples.

I'm still not convinced by the TIMIT experiments, now that I better understand them, since the F+M baseline is quite strong and very simple to run. It simply doesn't seem worthwhile to introduce all of this extra machinery for such a marginal improvement, but the experiment does serve the job of at least demonstrating an improvement over existing methods.

---

> ### Author Response · Authors · 2018-11-18
> **Response to AnonReviewer2 comments and suggestions**
>
> Comment on “low-resource supervised adaptation, Table 2”:
> To provide more baseline results on low-resource supervised adaptation, we ran additional experiments and replaced table 2 with bar plots in Figure 3. Baselines include classifier trained on low-resource target data, and including source data, with no adaptation. As shown in Figure 3, Augmented-Cyc algorithm outperforms FADA model and the two baselines.
>
> Comment on “comparing with existing works on low-resource unsupervised adaptation”:
> We added experiments on low-resource unsupervised adaptation to compare with CyCADA, and the results are shown in Figure 2. This experiment  investigates the effectiveness and robustness of using two cycle with semantic consistency enforced by auxiliary task loss, compared to CyCADA, where semantic consistency is enforced by reconstruction loss. As shown in Figure 2, CyCADA model fails to learn a good adaptation, where target domain contains few unsupervised data. Additionally, CyCADA model shows high instability in low-resource situation. Our model achieves more robust and better performance. We think this is attributed to proper use of source classifier to enforce consistency and robustness that we get by using two cycles (also shown in ablation study in Table 1).
>
> Comment on speech domain experiments:
>  We have edited the speech experiment section for more clarification. To mention some, “Adapted Male” is changed to “Male-> Female” to preserve consistency in notation. “All Data” refers to “Male+Female” with no adaption. CycleGAN results are added for"Female + Adapted Male," or "All Data + Adapted Male,”

---

> > ### Author Response · Authors · 2018-12-04
> > **More comparison with CyCADA**
> >
> > To provide more baselines, we added more comparisons with CyCADA model on low and high-resource unsupervised domain adaptation in updated Figure 2 and Table 2 in the following links
> >
> > Figure 2: https://bit.ly/2E812qf
> > Table 2:  https://bit.ly/2QwEogQ

---

### Official Review · AnonReviewer1 · 2018-11-08
**Interesting paper**

**Rating:** 8
**Confidence:** 2

**Review:**


I am putting "weak accept" because I think the paper addresses an important problem (domain adaptation) and has an interesting approach.  As the other reviewers pointed out, it's maybe not *super* novel.  But it's still interesting, and pretty readable for the most part.

I do question the statistical significance of the TIMIT experiments: TIMIT has a very tiny test set to start with, and by focusing on the female portion only you are further reducing the amount.

Small point: I don't think GANs are technically nonparametric, as the neural nets do have parameters.

I am a little skeptical that this method would have as general applicability or usefulness as the authors seem to think.  The reason is that, since the cycle constraint no longer exists, there is nothing to stop the network from just figuring out the class label of the input (say) image, and treating all the rest of the  information in that image as noise the same way a regular non-cyclic GAN would treat it.  Of course, one wouldn't expect a convolutional network to behave like this, but in theory it could happen in general cases.  This is just speculation though.  Personally I would have tended to accept the paper, but I'm not going to argue with the other reviewers, who are probably more familiar with GAN literature than me.

--
I am changing from "marginally above acceptance threshold" to "clear accept" after reading the response and thinking about the paper a bit more.  I acknowledge that the difference from previously published methods is not that large, but I still think it has value as it's getting quite close to being a practical method for generating fake training data for speech recognition.

---

> ### Author Response · Authors · 2018-11-18
> **Response to AnonReviewer1 comments and suggestions**
>
> - Comment on “statistical significance on TIMIT experiments”:
> We have chosen TIMIT dataset because of its inherent low-resource domain for different genders. As shown in Table 3, when using only Male speech for training the network, testing on female genders results in a large margin (11% on phoneme error recognition), compared to baseline. However by using only “Male->Female” data in training of proposed model, this gap can be reduced by ~10% for 124 voices in validation and 64 voices in test set for female domain.
>
>  - Comment on “Whether GAN’s are parametric or non-parametrics”:
> Here we refer to the classical parametric models for modeling data distribution. In this sense, the generator in GAN implicitly models the true distribution. Therefore, we categorize GAN as a non-parametric density estimation model since it does not assume any form of distribution.
>
> - Comment on general applicability of the proposed domain adaptation model:
> Since for any sample, whether target or source, there are two classifier in the cycle to preserve the class label information during transformation across domains, we believe that this implicit enforcement of content preservation will hold in broader applications. If the model is able to figure out which part is important for a certain class and ignore other parts, that is a desired behavior, since only those parts are important for the task in mind.

---

### Author Response · Authors · 2018-11-18
**General Response**

We appreciate all reviewers for providing insightful comments on technical aspect  of the proposed method.
Below, we summarize the revision briefly. Detailed responses to each reviewer/comment are followed based on each reviewer feedback.
- Additional experiment is performed to compare the performance of our model on low-resource unsupervised domain with CyCADA model (see Figure 2)
- Table 2 (low-resource supervised experiment) in the previous version is now replaced with Figure 3 (bar plot), where additional baselines are added, to emphasize the significance of domain adaptation in our model in comparison to state of the art models.
- Table 3 (Speech experiments): The updated acronyms in this table are now consistent with those in visual adaptation section.We further added additional CycleGAN results to provide mores baselines for comparison between models.
- The new title for the work is “Augmented Cyclic Adversarial Learning for Low Resource Domain Adaptation”
- The acronym for the proposed model is changed from “Augmented-Cyc” to “ACAL” and “Relaxed-Cyc” to “RCAL”
- In all experiments, Tables and Figures, number of samples are per class, whether supervised or unsupervised
- All changes are highlighted in the paper
- We have used the official CyCADA open-source code to reproduce its results and get the performance on low-resource unsupervised adaptation in Figure 2.
CyCADA code: https://github.com/jhoffman/cycada_release


Clarification on Novelty:
In this paper, we address the problem of domain adaptation for low resource situation in supervised, semi and unsupervised situations. We emphasize the necessity of using two cycle in tackling this problem. As evident from our experiments (see results in Table 1 and Figure 2), current one-cycle based models (such as CyCADA) or conventional two-cycle method (CycleGAN) fail in stable and good adaptation in low-resource situation.

---

> ### Author Response · Authors · 2018-12-01
> **Novelty: more clarification**
>
> We would like emphasize again that our focus in this paper is for low resource domain adaptation, and our main contribution and novelty is in the introduction of two cycles for low resource domain adaptation. From our experiments, it is clear that the introduction of an additional cycle is necessary to get robust performance in low resource settings, irrespective of whether learning is supervised or unsupervised. While this may seem like a subtle difference, the benefit of the additional cycle is clear, as shown in both our ablation study (Table 1) and the comparison between CyCADA and our method (Fig 2). Our intuition for this benefit is that conversion from both direction makes more training data available for the model, which results in more robust models. This improvement is both consistent and significant as compared to CycleGAN and other methods in various experiments under different settings, which shows the significance of the change.
>
> We agree that differences between our approach and previous approaches may be subtle; however, we argue that the key contribution of our work is both well motivated by the need to make efficient use of low resource data and is empirically supported by the various improvements we demonstrate.
>
> Updated Figure 2 and Table 2 are here,
> Figure 2: https://bit.ly/2E812qf
> Table 2:  https://bit.ly/2QwEogQ

---

### Meta-Review · Area_Chair1 · 2018-12-15
**Practically useful extension to CycleGAN in low-resource settings**

**Confidence:** 4
**Recommendation:** Accept (Poster)

**Metareview:**

The authors propose a method for low-resource domain adaptation where the number of examples available in the target domain are limited. The proposed method modifies the basic approach in a CycleGAN by augmenting it with a “content” (task-specific) loss, instead of the standard reconstruction error. The authors also demonstrate experimentally that it is important to enforce the loss in both directions (target → source and source --> target). Experiments are conducted on both supervised as well as unsupervised settings.
The main concern expressed by the reviewers relates to the novelty of the approach since it is a relatively straightforward extension of CycleGAN/CyCADA, but in the view of a majority of reviewers the work serves a useful contribution as a practical method for developing systems in low-resource conditions where it is feasible to label a few new instances. Although the reviewers were not unanimous in their recommendations, on balance in the view of the AC the work is a useful contribution with clear and detailed experiments in the revised version.